# One-shot Imitation Learning via Interaction Warping

**Ondrej Biza**[1], **Skye Thompson**[2], **Kishore Reddy Pagidi**[1], **Abhinav Kumar**[1],
**Elise van der Pol**[3,*], **Robin Walters**[1,*], **Thomas Kipf**[4,*]
**Jan-Willem van de Meent**[1,5,†], **Lawson L.S. Wong**[1,†], **Robert Platt**[1,†]
[1] Northeastern University, [2] Brown University, [3] Microsoft Research,
[4] Google DeepMind, [5] University of Amsterdam
[*] Equal contribution. [†] Equal advising.
`biza.o@northeastern.edu`

**Abstract:** Learning robot policies from few demonstrations is crucial in open-ended applications. We propose a new method, Interaction Warping, for one-shot learning SE(3) robotic manipulation policies. We infer the 3D mesh of each object in the environment using shape warping, a technique for aligning point clouds across object instances. Then, we represent manipulation actions as keypoints on objects, which can be warped with the shape of the object. We show successful one-shot imitation learning on three simulated and real-world object rearrangement tasks. We also demonstrate the ability of our method to predict object meshes and robot grasps in the wild. Webpage: https://shapewarping.github.io.

**Keywords:** 3D manipulation, imitation learning, shape warping

## 1 Introduction

In one-shot imitation learning, we are given a single demonstration of a desired manipulation behavior and we must find a policy that can reproduce the behavior in different situations. A classic example is the Mug Tree task, where a robot must grasp a mug and hang it on a tree by its handle. Given a single demonstration of grasping a mug and hanging it on a tree (top row of Figure 1), we want to obtain a policy that can successfully generalize across objects and poses, e.g. differently-shaped mugs and trees (bottom row of Figure 1). This presents two key challenges: First, the demonstration must generalize to novel object instances, e.g. different mugs. Second, the policy must reason in SE(3), rather than in SE(2) where the problem is much easier [1].

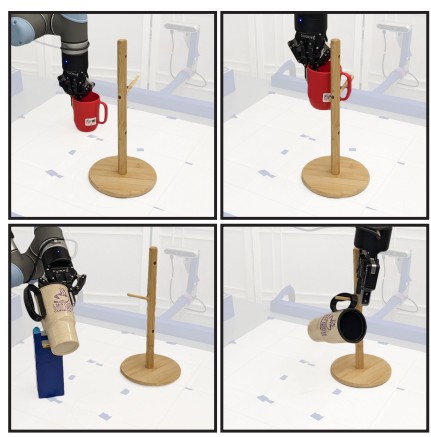

Figure 1: The Mug Tree task.

To be successful in SE(3) manipulation, it is generally necessary to bias the model significantly toward the object manipulation domains in question. One popular approach is to establish a correspondence between points on the surface of the objects in the demonstration(s) with the same points on the objects seen at test time. This approach is generally implemented using *keypoints*, point descriptors that encode the semantic location of the point on the surface of an object and transfer well between different novel object instances [2, 3, 4]. E.g., points on handles from different mugs should be assigned similar descriptors, thereby helping to correspond handles on different mug instances. A key challenge therefore becomes how to learn semantically meaningful keypoint descriptors. Early work used hand-coded feature labels [4]. More recent methods learn a category-level object descriptor models during a pre-training step using implicit object models [5] or point models [2].

7th Conference on Robot Learning (CoRL 2023), Atlanta, USA.

This paper proposes a different approach to the point correspondence problem based on Coherent Point Drift (CPD) [6], a point-cloud warping algorithm. We call this method *Interaction Warping*. Using CPD, we train a shape-completion model to register a novel in-category object instance to a canonical object model in which the task has been defined via a single demonstration. The canonical task can then be projected into scenes with novel in-category objects by registering the new objects to the canonical models. Our method has several advantages over the previous work mentioned above [2, 3, 4]. First, it performs better in terms of its ability to successfully perform a novel instance of a demonstrated task, both in simulation and on robotic hardware. Second, it requires an order-of-magnitude fewer object instances to train each new object category – tens of object instances rather than hundreds. Third, our method is agnostic to the use of neural networks – the approach presented is based on CPD and PCA models, though using neural networks is possible.

## 2    Related Work

We draw on prior works in shape warping [7, 8] and imitation learning via keypoints [4]. Shape warping uses non-rigid point cloud registration [9], a set of methods for aligning point clouds or meshes of objects, to transfer robot skills across objects of different shape. Our paper is the first to use shape warping to perform relational object re-arrangement and to handle objects in arbitrary poses. Second, keypoints are a state abstraction method that reduces objects states to the poses of a set of task-specific keypoints. We use keypoints to transfer robot actions. The novelty in our work is that our interaction points are found automatically and warped together with object shape.

**Few-shot Learning of Manipulation Policies:** Keypoint based methods have been used in few-shot learning of object re-arrangement [4, 10, 11]. These methods rely on human-annotated object keypoints. Follow-up works proposed learned keypoints for learning tool affordances [12, 13, 14] and for model-based RL [15]. A related idea is the learning of 2D [16] and 3D [5, 17, 18, 19] descriptor fields, which provide semantic embeddings for arbitrary points. A keypoint can then be matched across object instances using its embedding. We specifically compare to Simeonov et al. [5, 17] and show that our method requires fewer demonstrations. In separate lines of works, Pan et al. [2] (also included in our comparison) tackled object re-arrangement using cross-attention [20] between point clouds and Wen et al. [21] used pose estimation to solve precise object insertion.

**Shape Warping and Manipulation:** We use a learned model of in-class shape warping originally proposed by Rodriguez et al. [22]. This model was previously used to transfer object grasps [7, 23, 24] and parameters for skills such as pouring liquids [25, 8]. Our method jointly infers the shape and pose of an object; prior work assumed object pose to be either given [8] or detected using a neural pose detector [24]. Gradient descent on both the pose and the shape was previously used by Rodriguez et al. [7], Rodriguez and Behnke [23], but only to correct for minor deviations in pose. A second line of work transfers grasps by warping contact points [26, 27, 28, 29, 30, 22, 31, 32]. Finally, point-cloud warping has been used to manipulate deformable objects [33, 34].

## 3    Background

**Coherent Point Drift (CPD)**    Given two point clouds, $X^{(i)} \in \mathbb{R}^{n \times 3}$ and $X^{(j)} \in \mathbb{R}^{m \times 3}$, Coherent Point Drift (CPD) finds a displacement $W_{i \to j} \in \mathbb{R}^{n \times 3}$ of the points $X^{(i)}$ that brings them as close as possible (in an $L_2$ sense) to the points $X^{(j)}$ [6]. CPD is a non-rigid point-cloud registration method – each point in $X^{(i)}$ can be translated independently. CPD minimizes the following cost function,



Initial          Final

Figure 2: Coherent Point Drift warping.

$$J(W_{i \to j}) = -\sum_{k=1}^{m} \log \sum_{l=1}^{n} \exp\left(-\frac{1}{2\sigma^2}\left\|X_l^{(i)} + (W_{i \to j})_l - X_k^{(j)}\right\|\right) + \frac{\alpha}{2}\phi(W_{i \to j}), \qquad (1)$$

using expectation maximization over point correspondences and distances (see [6] for details). This can be viewed as fitting a Gaussian Mixture Model of $n$ components to the data $X^{(j)}$. Here, $\phi(W_{i \to j})$ is a prior on the point displacements that regularizes nearby points in $X^{(i)}$ to move coherently, preventing the assignment of arbitrary correspondences between points in $X^{(i)}$ and $X^{(j)}$.

**Generative Object Modeling Using CPD:** CPD can be used as part of a generative model for in-category object shapes as follows [7]. Assume that we are given a set of point clouds, $\{X^{(1)}, \ldots, X^{(K)}\}$, that describe $K$ object instances that all belong to a single category, e.g. a set of point clouds describing different mug instances. Each of these point clouds must be a full point cloud in the sense that it covers the entire object. Select a "canonical" object $X^{(C)}, C \in \{1, 2, ..., K\}$ and define a set of displacement matrices $W_{C \to i} = \text{CPD}(X^{(C)}, X^{(i)}), i \in \{1, 2, ..., K\}$. The choice of $C$ is arbitrary, but we heuristically choose the $C$ that is the most representative (Appendix A.2). Now, we calculate a low rank approximation of the space of object-shape deformations using PCA. For each matrix $W_{C \to i} \in \mathbb{R}^{n \times 3}$, let $\bar{W}_{C \to i} \in \mathbb{R}^{3n \times 1}$ denote the flattened version. We form the $3n \times K$ data matrix $\bar{W}_C = \left[ \bar{W}_{C \to 1}, \ldots, \bar{W}_{C \to K} \right]$ and calculate the $d$-dimensional PCA projection matrix $W \in \mathbb{R}^{3n \times d}$. This allows us to approximate novel in-category objects using a low-dimensional latent vector $v_{\text{novel}} \in \mathbb{R}^d$, which can be used to compute a point cloud

$$Y = X^{(C)} + \text{Reshape}(W v_{\text{novel}}), \tag{2}$$

where the Reshape operator casts back to an $n \times 3$ matrix.

**Shape Completion From Partial Point Clouds:** In practice, we want to be able to approximate complete point clouds for objects for which we only have a partial view [8]. This can be accomplished using the generative model by solving for

$$\mathcal{L}(Y) = \mathcal{D}(Y, X^{(\text{partial})}), \tag{3}$$

using gradient descent on $v$. Essentially, we are solving for the latent vector that gives us a reconstruction closest to the observed points. To account for the partial view, Thompson et al. [8] use the one-sided Chamfer distance [35],

$$\mathcal{D}\left(X^{(i)}, X^{(j)}\right) = \frac{1}{m} \sum_{k=1}^{m} \min_{l \in \{1, ..., n\}} \left\| X_l^{(i)} - X_k^{(j)} \right\|_2. \tag{4}$$

Note that $X^{(i)} \in \mathbb{R}^{n \times 3}$ and $X^{(j)} \in \mathbb{R}^{m \times 3}$ do not need to have the same number of points ($n \neq m$).

## 4 Interaction Warping

This section describes **Interaction Warping (IW)**, our proposed imitation method (Figure 3). We assume that we have first trained a set of category-level generative object models of the form described in Section 3. Then, given a single demonstration of a desired manipulation activity, we

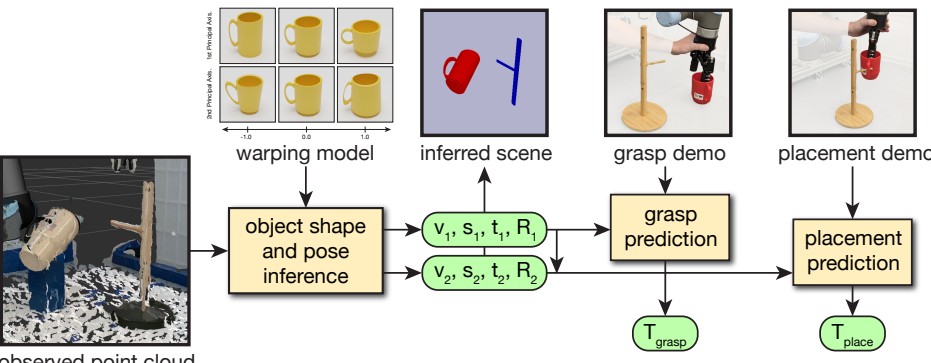

Figure 3: Interaction Warping pipeline for predicting grasp and placement poses from point clouds.

detect the objects in the demonstration using off-the-shelf models. For each object in the demonstration that matches a previously trained generative model, we fit the model to the object in order to get the pose and completed shape of the object (Section 4.1 and 4.2). Next, we identify *interaction points* on pairs of objects that interact and corresponding those points with the matching points in the canonical object models. Finally, we reproduce the demonstration in a new scene with novel in-category object instances by projecting the demonstrated interaction points onto the completed object instances in the new scene (Section 4.3).

## 4.1 Joint Shape and Pose Inference

In order to manipulate objects in $\mathrm{SE}(3)$, we want to jointly infer the pose and shape of an object represented by a point cloud $\mathrm{X}^{(\mathrm{partial})}$. To do so, we warp and transform point cloud $Y \in \mathbb{R}^{n \times 3}$ to minimize a loss function,

$$\mathcal{L}(Y) = \mathcal{D}(Y, \mathrm{X}^{(\mathrm{partial})}) + \beta \max_k \|Y_k\|_2^2, \tag{5}$$

which is akin to Equation 3 with the addition of the second term, a regularizer on the size of the decoded object. Our implementation regularizes the object to fit into the smallest possible ball. The main reason for the regularizer is to prevent large predicted meshes in real-world experiments, which might make it impossible to find collision-free motion plans.

We parameterize $Y$ as a warped, scaled, rotated and translated canonical point cloud,

$$Y = [\underbrace{(\mathrm{X}^{(C)} + \mathrm{Reshape}(Wv))}_{\text{Equation 2}} \odot s] R^T + t. \tag{6}$$

Here, $\mathrm{X}^{(C)}$ is a canonical point cloud and $v \in \mathbb{R}^d$ parameterizes a warped shape (as described in Section 3), $s \in \mathbb{R}^3$ represents scale, $R \in \mathrm{SO}(3)$ is a rotation matrix and $t \in \mathbb{R}^3$ represents translation. We treat $s$ and $t$ as row vectors in this equation.

We directly optimize $\mathcal{L}$ with respect to $v, s$ and $t$ using the Adam optimizer [36]. We parameterize $R$ using $\hat{R} \in \mathbb{R}^{2 \times 3}$, an arbitrary matrix, and perform Gram-Schmidt orthogonalization (Algorithm 5) to compute a valid rotation matrix $R$. This parameterization has been shown to enable stable learning of rotation matrices [37, 38]. We run the optimization with many initial random restarts, please see Appendix A.4 for further details. The inferred $v, s$ represent the shape of the object captured by $\mathrm{X}^{(\mathrm{partial})}$ and $R, t$ represent its pose.

## 4.2 From Point Clouds to Meshes

We infer the shape and pose of objects by warping point clouds. But, we need object meshes to perform collision checking for finding contacts between objects and motion planning (Section 4.3). We propose a simple approach for recovering the mesh of a warped object based on the vertices and faces of the canonical object.

First, we warp the vertices of the canonical object. To do so, the vertices need to be a part of $\mathrm{X}^{(C)}$ because our model only knows how to warp points in $\mathrm{X}^{(C)}$ (Section 3). However, these vertices (extracted from meshes made by people) are usually very biased (e.g. 90% of the vertices might be in the handle of a mug), which results in learned warps that ignore some parts of the object. Second, we add points to $\mathrm{X}^{(C)}$ that are randomly sampled on the surface of the canonical mesh. $\mathrm{X}^{(C)}$ is then composed of approximately the same number of mesh vertices and random surface samples, leading to a better learned warping. We construct $\mathrm{X}^{(C)}$ such that the first $V$ points are the vertices; note that the ordering of points in $\mathrm{X}^{(C)}$ does not change as it is warped.

Given a warped, rotated and translated point cloud $Y$ (Equation 6), the first $V$ points are the warped mesh vertices. We combine them with the faces of the canonical object to create a warped mesh $M$. Faces are represented as triples of vertices and these stay the same across object warps.

## 4.3 Transferring Robot Actions via Interaction Points

Consider the example of a point cloud of a mug $Y$ that is warped using Equation 6. We can select any point $Y_i$ and track it as the mug changes its shape and pose. For example, if the point lies on the handle of the mug, we can use it to align handles of mugs of different shapes and sizes. That can, in turn, facilitate the transfer of manipulation policies across mugs. The key question is how to find the points $Y_i$ relevant to a particular task. We call these *interaction points*.

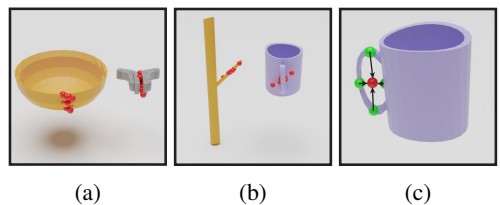

(a)       (b)       (c)

Figure 4: (a) Contacts between a gripper and a bowl extracted from a demonstration. (b) Nearby points between a mug and a tree extracted from a demonstration of hanging the mug on the tree. (c) A virtual point (red) representing the branch of the tree intersecting the handle of the mug. The red point is anchored to the mug using k nearest neighbors on the mug (four are shown in green).

**Grasp Interaction Points:** We define the grasp interaction points as the pairs of contact points between the gripper and the object at the point of grasp. Let $Y^{(A)}$ and $M^{(A)}$ be the point cloud and mesh respectively for the grasped object inferred by our method (Section 4.1, 4.2). Let $M^{(G)}$ be a mesh of our gripper and $T_G$ the pose of the grasp. We use `pybullet` collision checking to find $P$ pairs of contact points $(p_j^{(A)}, p_j^{(G)})_{j=1}^P$, where $p_j^{(A)}$ is on the surface of $M^{(A)}$ and $p_j^{(G)}$ is on the surface of $M^{(G)}$ in pose $T_G$ (Figure 4a). We want to warp points $p_j^{(A)}$ onto a different shape, but our model only knows how to warp points in $Y^{(A)}$. Therefore, we find a set of indices $I_G = \{i_1, ..., i_P\}$, where $Y_{i_j}^{(A)}$ is the nearest neighbor of $p_j^{(A)}$.

**Transferring Grasps:** In a new scene, we infer the point cloud of the new object $Y^{(A')}$ (Eq. 6). We solve for the new grasp as the optimal transformation $T_G^*$ that aligns the pairs of points $(Y_{i_j}^{(A')}, p_j^{(G)}), j \in \{1, ..., P\}, i_j \in I_G$. Here, $Y_{i_j}^{(A')}$ are the contact points from the demonstration warped to a new object instance. Note that there is a correspondence between the points in $Y^{(A)}$ and $Y^{(A')}$; shape warping does not change their order. We predict the grasp $T_G^*$ (Figure 5a) that minimizes the pairwise distances analytically using an algorithm from Horn et al. [39].

**Placement Interaction Points:** For placement actions, we look at two objects being placed in relation to each other, such as a mug being placed on a mug-tree. Here, we define interaction points as pairs of *nearby points* between the two object, a generalization of contact points. We use nearby points so that the two objects do not have to make contact in the demonstration; e.g., the mug might not be touching the tree before it is released from the gripper. Similarly, the demonstration of an object being dropped into a container might not include contacts.

Let $Y^{(A)}$ and $Y^{(B)}$ be the inferred point clouds of the two objects. We capture the original point clouds from a demonstration right before the robot opens its gripper. We find pairs of nearby points with $L_2$ distance below $\delta$, $\{(p^{(A)} \in Y^{(A)}, p^{(B)} \in Y^{(B)}) : \left\| p^{(A)} - p^{(B)} \right\| < \delta\}$. Since there might be tens of thousands of these pairs, we find a representative sample using farthest point sampling [40]. We record the indices of points $p_j^{(B)}$ in $Y^{(B)}$ as $I_P = \{i_1, i_2, ..., i_P\}$.

We further add $p_j^{(B)}$ as **virtual points** into $Y^{(A)}$ – this idea is illustrated in Figure 4 (b) and (c). For example, we wish to solve for a pose that places a mug on a tree, such that the branch of the tree intersects the mug's handle. But, there is no point in the middle of the mug's handle that we can use. Hence, we add the nearby points $p_j^{(B)}$ (e.g. points on the branch of the tree) as virtual points $q_j^{(A)}$ to $Y^{(A)}$. We anchor $q_j^{(A)}$ using L-nearest-neighbors so it warps together with $Y^{(A)}$. Specifically, for each point $p_j^{(B)}$ we find $L$ nearest neighbors $(n_{j,1}, ..., n_{j,L})$ in $Y^{(A)}$ and anchor $q_j^{(A)}$ as follows,

$$q_j^{(A)} = \frac{1}{L} \sum_{k=1}^{L} Y_{n_{j,k}}^{(A)} + \underbrace{(p_j^{(B)} - Y_{n_{j,k}}^{(A)})}_{\Delta_{j,k}} = p_j^{(B)}. \tag{7}$$

To transfer the placement, we save the neighbor indices $n_{j,k}$ and the neighbor displacements $\Delta_{j,k}$.

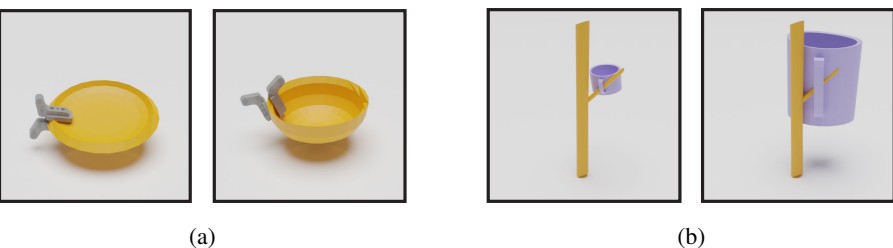

(a)                                                                    (b)

Figure 5: Predicting grasps using interaction point warping. (a) the predicted grasp for a bowl/plate changes based on the curvature of the object. (b) the placement of a mug on a mug tree changes as the mug grows larger so that the branch of the tree is in the middle of the handle.

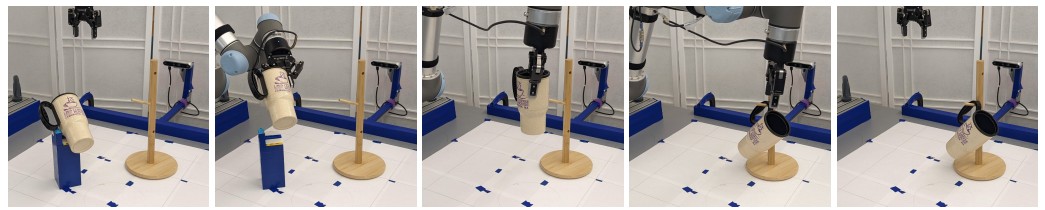

Figure 6: Example of an episode of putting a mug on a tree starting from a tilted mug pose.

**Transferring Placements:** We infer the point clouds of the pair of new objects $Y^{(A')}$ and $Y^{(B')}$. We calculate the positions of the virtual points with respect to the warped nearest neighbors,

$$q_j^{(A')} = \frac{1}{L} \sum_{k=1}^{L} Y_{n_{j,k}}^{(A')} + \Delta_{j,k}. \tag{8}$$

We then construct pairs of points $(q_j^{(A')}, Y_{i_j}^{(B')}), j \in \{1, ..., P\}, i_j \in I_P$ and find the optimal transformation of the first object $T_P^*$ that minimizes the distance between the point pairs. Since we know how we picked up the first object, we can transform $T_P^*$ into the coordinate frame of the robot hand and execute the action of placing object $A'$ onto object $B'$ (Figure 5b).

## 5 Experiments

We evaluate both the perception and imitation learning capabilities of Interaction Warping. In Section 5.1, we perform three object re-arrangement tasks with previously unseen objects both in simulation and on a physical robot. In Section 5.2, we show our system is capable of proposing grasps in a cluttered kitchen setting from a single RGB-D image.

We use ShapeNet [41] for per-category (mug, bowl, bottle and box) object pre-training (required by our method and all baselines). We use synthetic mug-tree meshes provided by [17]. Our method (IW) uses 10 training example per class, whereas all baselines use 200 examples. The training meshes are all aligned in a canonical pose.

### 5.1 Object Re-arrangement

**Setup:** We use an open-source simulated environment with three tasks: mug on a mug-tree, bowl on a mug and a bottle in a container [17]. Given a segmented point cloud of the initial scene, the goal is to predict the pose of the child object relative to the parent object (e.g. the mug relative to the mug-tree). A successful action places the object on a rack / in a container so that it does not fall down, but also does not clip within the rack / container. The simulation does not test grasp prediction. The three tasks are demonstrated with objects unseen during pre-training. We described how our method (IW) uses a single demonstration in Section 4.3; to use multiple demonstration, IW uses training prediction error to select the most informative one (Appendix A.5).

| Method | # Demo | # Train. Meshes | Mug on Tree | | Bowl on Mug | | Bottle in Container | |
|---|---|---|---|---|---|---|---|---|
| | | | Upright | Arbitrary | Upright | Arbitrary | Upright | Arbitrary |
| R-NDF [17] | 1 | 200 | 60.0 | 51.0 | 69.0 | 68.0 | 19.0 | 8.0 |
| TAX-Pose [2] | 1 | 200 | 61.0 | 41.0 | 16.0 | 9.0 | 4.0 | 1.0 |
| **IW (Ours)** | 1 | 10 | **86.0** | **83.0** | **82.0** | **84.0** | **62.0** | **60.0** |
| R-NDF [17] | 5 | 200 | 88.0 | **89.0** | 53.0 | 46.0 | 78.0 | 47.0 |
| TAX-Pose [2] | 5 | 200 | 82.0 | 51.0 | 29.0 | 14.0 | 6.0 | 2.0 |
| **IW (Ours)** | 5 | 10 | **90.0** | 87.0 | **75.0** | **77.0** | **79.0** | **79.0** |
| R-NDF [17] | 10 | 200 | 71.0 | 70.0 | 69.0 | 60.0 | **81.0** | 59.0 |
| TAX-Pose [2] | 10 | 200 | 82.0 | 52.0 | 20.0 | 20.0 | 2.0 | 1.0 |
| **IW (Ours)** | 10 | 10 | **88.0** | **88.0** | **83.0** | **86.0** | 70.0 | **83.0** |

Table 1: Success rates of predicted target poses of objects in simulation. Upright and Arbitrary refer to the starting pose of the manipulated object. Measured over 100 trials with unseen object pairs.

| Method | Mug on Tree | | Bowl on Mug | | Bottle in Container | | Mean | |
|---|---|---|---|---|---|---|---|---|
| | Pick | Pick&Place | Pick | Pick&Place | Pick | Pick&Place | Pick | Pick&Place |
| NDF[1] [5] | 93.3 | 26.7 | 75.0 | 33.3 | 20.0 | 6.7 | 62.8 | 22.2 |
| R-NDF [17] | 64.0 | 12.0 | 37.5 | 37.5 | 26.7 | 20.0 | 42.7 | 23.2 |
| **IW (Ours)** | **96.0** | **92.0** | **87.5** | **83.3** | **86.7** | **83.3** | **90.1** | **86.2** |

Table 2: Success rates of real-world pick-and-place experiments with a single demonstration. The manipulated object (e.g. a mug) starts in an arbitrary pose (we use a stand to get a range of poses) and the target object (e.g. a mug-tree) starts in an arbitrary upright pose. [1]The target object (e.g. the mug tree) is in a fixed pose for this experiment, as NDF does not handle target object variation. Each entry is measured over 25 - 30 trials with unseen object pairs.

In our real-world experiment, we perform both grasps and placements based on a single demonstration. We capture a fused point cloud using three RGB-D cameras. We use point-cloud clustering and heuristics to detect objects in the real-world scenes (details in Appendix B.1) and perform motion planning with collision checking based on the meshes predicted by our method. We evaluate the ability of each method to pick and place unseen objects with a varying shape and pose (Figure 8). We provide a single demonstration for each task by teleoperating the physical robot. We do not have access to the CAD models of objects used in the real-world experiment.

**Result:** We find that our method (IW) generally outperforms R-NDF [5] and TAX-Pose [2] on the simulated relational-placement prediction tasks (Table 1) with 20 times fewer training objects. We chose these two baselines as recent state-of-the-art SE(3) few-shot learning methods. IW can usually predict with above 80% success rate even with 1 demo, whereas R-NDF and TAX-Pose can only occasionally do so with 5+ demos, and often fail to reach 80% success rate at all. We use an open-source implementation of R-NDF provided by the authors [42], which differs in performance from the results reported in [17]. TAX-Pose struggles with precise object placements in the bowl on mug and bottle in box tasks; it often places the pair of objects inside one another. Occasionally, adding more demonstrations decreases the success rate because some demonstrations are of low quality (e.g. using decorative mugs with strange shapes).

In real-world pick and place experiments, we demonstrate the ability of IW to solve the three object re-arrangement tasks – mug on tree, bowl on mug and bottle in box – with unseen objects (Figure 8) and variation in the starting pose of objects (Table 2). We find that NDF and R-NDF [5, 17] struggle with the partial and noisy real-world point clouds. This often results in both the pick and place actions being too imprecise to successfully solve the task. Pre-training (R-)NDF on real-world point clouds could help, but note that IW was also pre-trained on simulated point clouds. We find that the warping of canonical objects is more robust to noisy and occluded point clouds. We show an example episode of placing a mug on a tree in Figure 6.

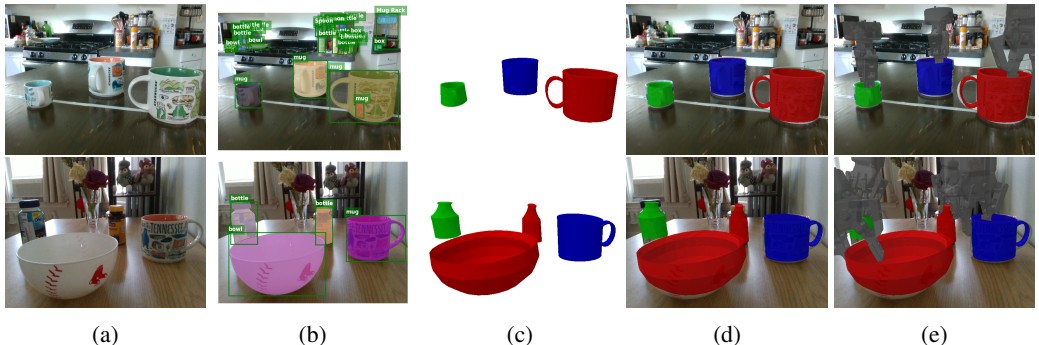

|     (a)     |     (b)     |     (c)     |     (d)     |     (e)     |

Figure 7: Grasp prediction in the wild: (a) an RGB-D (depth not shown) image, (b) open-vocabulary object detection and segmentation using Detic [43] and Segment Anything [44], (c) object meshes predicted by our method based on segmented point clouds (we filter out distant and small objects), (d) meshes projected into the original image, (e) grasps predicted by Interaction Warping projected into the original image. Figure 9 has additional examples.

We use the meshes predicted by IW to perform collision checking during motion planning. We do not perform collision checking (other than to avoid contact with the table) when using (R-)NDF as these methods do not predict object meshes, but failures due to a collision between the robot and one of the object were infrequent in real-world (R-)NDF trials.

## 5.2 Grasp Prediction in the Wild

**Setup:** In this experiment, we show that we can combine our method with a state-of-the-art object detection and segmentation pipeline to predict object meshes and robot grasps from a single RGB-D image. We use an open-vocabulary object detector Detic [43] to predict bounding boxes for common household objects and Segment Anything [44] to predict segmentation masks within these bounding boxes. We turn the predicted RGB-D images into point clouds and use our shape warping model to predict a mesh for each object. Finally, we use interaction warping to predict a robot grasp based on a single demonstration per each object class (details in Appendix B.2).

**Result:** We show the results for two example scenes in Figure 7 and 9. Our perception pipeline can successfully detect objects in images with cluttered backgrounds. Our warping algorithm accounts for the variation in the shape and size of objects and our interaction warping algorithm can generalize the demonstrated grasps to the novel objects.

## 6 Limitations and Conclusion

We introduced Interaction Warping, a method for one-shot learning of SE(3) robotic manipulation policies. We demonstrated that warping of shapes and interaction points leads to successful one-shot learning of object re-arrangement policies. We also showed that we can use open-vocabulary detection and segmentation models to detect objects in the wild and predict their meshes and grasps.

**Limitations:** Our method requires segmented point clouds of objects. We demonstrated a real-world object detection pipeline in Section 5.2, but it can be difficult to capture clean point clouds aligned with image-based segmentations. The joint inference of shape and pose of an object takes around 25 seconds per object on an NVIDIA RTX 2080 Ti GPU. Future work could train an additional neural network to amortize the inference, or to predict favorable initialization. We use a PCA model of shape warps for simplicity; this model cannot capture the details of objects, such as the detailed shape of the head of a bottle. A model with higher capacity should be used for tasks that require high precision. Finally, our predicted policy is fully determined by the shape warping model and a single demonstration; our method does not learn from its failures, but it is fully differentiable.

**Acknowledgments**

This work was supported in part by NSF 1724191, NSF 1750649, NSF 1763878, NSF 1901117, NSF 2107256, NSF 2134178, NASA 80NSSC19K1474 and NSF GRFP awarded to Skye Thompson. We would like the thank the CoRL reviewers and area chair for their feedback.

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

# A Method Details

We included the code for both our simulated and real-world experiments for reference. Please find it in the supplementary material under `iw_code`. Algorithms 1 and 2 describe our warp learning and inference.

---

**Algorithm 1** Warp Learning

---

**Input:** Meshes of $K$ example object instances $\{\mathrm{obj}_1, \mathrm{obj}_2, ..., \mathrm{obj}_K\}$.
**Output:** Canonical point cloud, vertices and faces and a latent space of warps.
**Parameters:** Smoothness of CPD warping $\alpha$ and number of PCA components $L$.

1: $\mathrm{PCD} = \langle \mathrm{SampleS}(\mathrm{obj}_i) \rangle_{i=1}^K$.         ▷ Sample a small point cloud per object (Appendix A.1).
2: $C = \mathrm{SelectCanonical}(\mathrm{PCD})$.         ▷ Select a canonical object with index $C$ (Appendix A.2).
3: $\mathrm{canon} = \mathrm{Concat}(\mathrm{obj}_C.\mathrm{vertices}, \mathrm{SampleL}(\mathrm{obj}_C))$.    ▷ Use both vertices and surface samples.
4: **for** $i \in \{1, 2, ..., K\}, i \neq C$ **do**
5:     $W_{C \rightarrow i} = \mathrm{CPD}(\mathrm{canon}, \mathrm{PCD}_i, \alpha)$.         ▷ Coherent Point Drift warping (Section 3).
6: **end for**
7: $D_W = \{\mathrm{Flatten}(W_{C \rightarrow i})\}_{i=1, i \neq C}^K$.         ▷ Dataset of displacements of canon.
8: $\mathrm{PCA} = \mathrm{FitPCA}(D_W, \mathrm{n\_components} = L)$. ▷ Learn a latent space of canonical object warps.
9: **return** $\mathrm{Canon}(\mathrm{points} = \mathrm{canon}, \mathrm{vertices} = \mathrm{obj}_C.\mathrm{vertices}, \mathrm{faces} = \mathrm{obj}_C.\mathrm{faces}), \mathrm{PCA}$.

---

**Algorithm 2** Warp Inference and Mesh Reconstruction

---

**Input:** Observed point cloud $\mathrm{pcd}$, canonical object $\mathrm{canon}$ and latent space PCA.
**Output:** Predicted latent shape $v$ and pose $T$.
**Parameters:** Number of random starts $S$, number of gradient descent steps $T$, learning rate $\eta$ and object size regularization $\beta$.

1: $t_g = \frac{1}{|\mathrm{pcd}|} \sum_{i=1}^{|\mathrm{pcd}|} \mathrm{pcd}_i$.
2: $\mathrm{pcd} = \mathrm{pcd} - t_g$.         ▷ Center the point cloud.
3: **for** $i = 1$ **to** $S$ **do**
4:     $R_{\mathrm{init}} = $ Random initial 3D rotation matrix.
5:     Initialize $v = \begin{pmatrix} 0 & 0 & ... & 0 \end{pmatrix}, s = \begin{pmatrix} 1 & 1 & 1 \end{pmatrix}, t_l = \begin{pmatrix} 0 & 0 & 0 \end{pmatrix}, \hat{R} = \begin{pmatrix} 1 & 0 & 0 \\ 0 & 1 & 0 \end{pmatrix}$.
6:     Initialize Adam [36] with parameters $v, s, t_l, r$ and learning rate $\eta$.
7:     **for** $j = 1$ **to** $T$ **do**
8:         $\delta = \mathrm{Reshape}(Wv)$.
9:         $X = \mathrm{canon.points} + \delta$.         ▷ Warped canonical point cloud.
10:         $R = \mathrm{GramSchmidt}(\hat{R})$.
11:         $X = (X \odot s)R_{\mathrm{init}}^T R^T + t_l$.         ▷ Scaled, rotated and translated point cloud.
12:         $\mathcal{L} = \frac{1}{|\mathrm{pcd}|} \sum_k^{|\mathrm{pcd}|} \min_l^{|X|} \|\mathrm{pcd}_k - X_l\|_2^2$.         ▷ One-sided Chamfer distance.
13:         $\mathcal{L} = \mathcal{L} + \beta \max_l^{|X|} \|X_l\|_2^2$.         ▷ Object size regularization.
14:         Take a gradient descent step to minimize $\mathcal{L}$ using Adam.
15:     **end for**
16: **end for**
17: Find parameters $v^*, s^*, t_l^*, R_{\mathrm{init}}^*, R^*$ with the lowest final loss across $i \in \{1, 2, ..., S\}$.
18: $X = \mathrm{canon.points} + \mathrm{Reshape}(Wv^*)$.
19: $X = (X \odot s^*)(R_{\mathrm{init}}^*)^T (R^*)^T + t_l^* + t_g$.     ▷ Complete point cloud in workspace coordinates.
20: $\mathrm{vertices} = \langle X_1, X_2, ..., X_{|\mathrm{canon.vertices}|} \rangle$.         ▷ First $|\mathrm{canon.vertices}|$ points of $X$ are vertices.
21: **return** $\mathrm{Mesh}(\mathrm{vertices} = \mathrm{vertices}, \mathrm{faces} = \mathrm{canon.faces})$.         ▷ Warped mesh.

---

## A.1 Point Cloud Sampling

We use `trimesh`[1] to sample the surface of object meshes. The function `trimesh.sample.sample_surface_even` samples a specified number of points and then rejects points that are too close together. We sample 2k points for small point clouds (SampleS) and 10k point for large point clouds (SampleL).

## A.2 Canonical Object Selection

Among the $K$ example objects, we would like to find the one that is the easiest to warp to the other objects. For example, if we have ten examples of mugs, but only one mug has a square handle, we should not choose it as it might be difficult to warp it to conform to the round handles of the other nine mugs. We use Algorithm 3, which computes $K * K - 1$ warps and picks the object that warps to the other $K - 1$ objects with the lowest Chamfer distance. We also note an alternative and computationally cheaper algorithm from Thompson et al. [8], Algorithm 4. This algorithm simply finds the object that is the most similar to the other $K - 1$ objects without any warping.

---

**Algorithm 3** Exhaustive Canonical Object Selection

---

**Input:** Point clouds of $K$ training objects $\langle X^{(1)}, X^{(2)}, ..., X^{(K)} \rangle$.
**Output:** Index of the canonical object.

1: **for** $i = 1$ **to** $K$ **do**
2:     **for** $j = 1$ **to** $K, j \neq i$ **do**
3:         $W_{i \rightarrow j} = \text{CPD}(X^{(i)}, X^{(j)})$         ▷ Warp point cloud $i$ to point cloud $j$.
4:         $C_{i,j} = \frac{1}{|X^{(j)}|} \sum_{k=1}^{|X^{(j)}|} \min_{l=1}^{|X^{(i)}|} \left\| X_k^{(j)} - (X^{(i)} + W_{i \rightarrow j})_l \right\|_2^2$
5:     **end for**
6: **end for**
7: **for** $i = 1$ **to** $K$ **do**
8:     $C_i = \sum_{j=1, j \neq i}^{K} C_{i,j}$         ▷ Cumulative cost of point cloud $i$ warps.
9: **end for**
10: **return** $\arg \min_{i=1}^{K} C_i$         ▷ Pick point cloud that is the easiest to warp.

---

**Algorithm 4** Approximate Canonical Object Selection [8]

---

**Input:** Point clouds of $K$ training objects $\langle X^{(1)}, X^{(2)}, ..., X^{(K)} \rangle$.
**Output:** Index of the canonical object.

1: **for** $i = 1$ **to** $K$ **do**
2:     **for** $j = 1$ **to** $K, j \neq i$ **do**
3:         $C_{i,j} = \frac{1}{|X^{(j)}|} \sum_{k=1}^{|X^{(j)}|} \min_{l=1}^{|X^{(i)}|} \left\| X_k^{(j)} - X_l^{(i)} \right\|_2^2$
4:     **end for**
5: **end for**
6: **for** $i = 1$ **to** $K$ **do**
7:     $C_i = \sum_{j=1, j \neq i}^{K} C_{i,j}$
8: **end for**
9: **return** $\arg \min_{i=1}^{K} C_i$

---

## A.3 Gram-Schmidt Orthogonalization

We compute a rotation matrix from two 3D vectors using Algorithm 5 [38].

---

[1]https://github.com/mikedh/trimesh

---

**Algorithm 5** Gram-Schmidt Orthogonalization

---

**Input:** 3D vectors $u$ and $v$.
**Output:** Rotation matrix.

1: $u' = u/\|u\|$
2: $v' = \frac{v-(u' \cdot v)u'}{\|v-(u' \cdot v)u'\|}$
3: $w' = u' \times v'$
4: **return** $\text{Stack}(u', v', w')$

---

## A.4 Shape and Pose Inference Details

The point clouds $Y \in \mathrm{R}^{n \times 3}$ starts in its canonical form with the latent shape $v$ equal to zero. We set the initial scale $s$ to one, translation $t$ to zero and rotation $\hat{R}$ to identity,

$$v = \underbrace{\begin{pmatrix} 0 & 0 & ... & 0 \end{pmatrix}}_{d}, \quad s = \begin{pmatrix} 1 & 1 & 1 \end{pmatrix}, \quad t = \begin{pmatrix} 0 & 0 & 0 \end{pmatrix}, \quad \hat{R} = \begin{pmatrix} 1 & 0 & 0 \\ 0 & 1 & 0 \end{pmatrix}. \tag{9}$$

$\hat{R}$ is then transformed into $R \in \mathrm{SO}(3)$ using Algorithm 5. We minimize $\mathcal{L}$ with respect to $v, s, t$ and $\hat{R}$ using the Adam optimizer [36] with learning rate $10^{-2}$ for 100 steps. We set $\beta = 10^{-2}$. We found the optimization process is prone to getting stuck in local minima; e.g., instead of aligning the handle of the decoded mug with the observed point cloud, the optimizer might change the shape of the decoded mug to hide its handle. Hence, we restart the process with many different random initial rotations and pick the solution with the lowest loss function. Further, we randomly subsample $Y$ to 1k points at each gradient descent step – this allows us to run 12 random starting orientations at once on an NVIDIA RTX 2080Ti GPU.

## A.5 Using Multiple Demonstrations

Our method transfers grasps and placements from a single demonstration, but in our simulated experiment, we have access to multiple demonstrations. We implement a simple heuristic for choosing the demonstration that fits our method the best: we make a prediction of the relational object placement from the initial state of each demonstration and select the demonstration where our prediction is closest to the demonstrated placement. The intuition is that we are choosing the demonstration where our method was able to warp the objects with the highest accuracy (leading to the best placement prediction). This is especially useful in filtering out demonstrations with strangely shaped objects.

# B Experiment Details

## B.1 Object re-arrangement on a physical robot

We use a UR5 robotic arm with a Robotiq gripper. We capture the point cloud using three RealSense D455 camera with extrinsics calibrated to the robot. For motion planning, we use MoveIt with ROS1. To segment the objects, we use DBSCAN to cluster the point clouds and simple heuristics (e.g. height, width) to detect the object class.

## B.2 Grasp prediction in the wild

We use a single RealSense D435 RGB-D camera. Our goal is to be able to demonstrate any task in the real world without having to re-train our perception pipeline. Therefore, we chose an open-vocabulary object detection model Detic [43], which is able to detect object based on natural language descriptions. We used the following classes: "cup", "bowl", "mug", "bottle", "cardboard", "box", "Tripod", "Baseball bat", "Lamp", "Mug Rack", "Plate", "Toaster" and "Spoon". We use

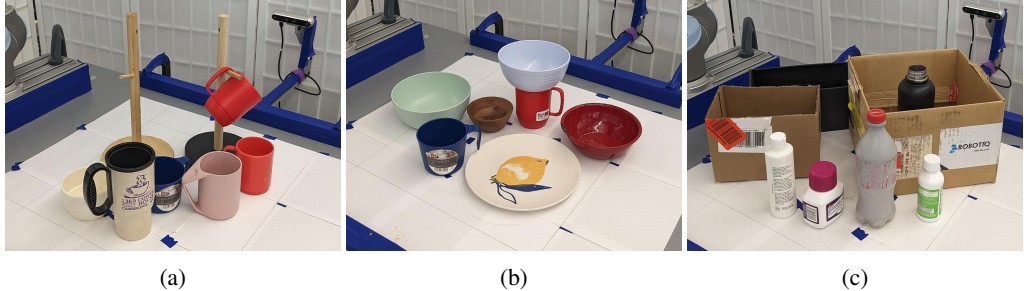

(a)       (b)       (c)

Figure 8: Objects used for the real-world tasks: (a) mug on tree, (b) bowl (or plate) on mug and (c) bottle in box. We use a single pair of objects to generate demonstrations and test on novel objects.

the predicted bounding boxes from Detic to condition a Segment Anything model [44] to get accurate class-agnostic segmentation masks. Both Detic[2] and Segment Anything[3] come with several pre-trained models and we used the largest available. Finally, we select the pixels within each segmentation mask and use the depth information from our depth camera to create a per-object point cloud. We use DBSCAN to clouster the point cloud and filter out outlier points. Then, we perform mesh warping and interaction warping to predict object meshes and grasps.

Previously, we experimented with Mask R-CNN [45] and Mask2Former [46] trained on standard segmentation datasets, such as COCO [47] and ADE20k [48]. We found that these dataset lack the wide range of object classes we would see in a household environment and that the trained models struggle with out-of-distribution viewing angles, such as looking from a steep top-down angle. We also experimented with an open-vocabulary object detection model OWL-ViT [49] and found it to be sensitive to scene clutter and the viewing angle.

## C  Additional Results

**Training and inference times:** We measure the training and inference times of TAX-Pose, R-NDF and IW (Table 3). Both R-NDF and IW take tens of seconds to either perceive the environment or to predict an action. This is because both of these methods use gradient descent with many random restarts for inference. On the other hand, TAX-Pose performs inference in a fraction of second but requires around 16 hours of training for each task. Neither R-NDF nor IW require task-specific training. We do not include the time it takes to perform pre-training for each class of objects, which is required by all three methods, because we used checkpoints provided by the authors of TAX-Pose and R-NDF.

**Additional real-world grasp predictions:** We include additional examples of real-world object segmentation, mesh prediction and grasp prediction in Figure 9.

## D  Limitations

**Limitations of shape warping:** Shape warping works well when we can smoothly warp shapes between object instances, but it would struggle with a changing number of object parts. For example, if we had a set of mug trees that have between one and six branches, shape warping would pick one of these trees as the canonical object and it would not be able to change the number of branches in the canonical tree.

Further, many object-oriented point cloud based methods (like IW and NDF) are limited by the receptive field of the point cloud they model. For example, if we wanted to perform a cooking task, both of these methods would not be able to model the entire kitchen aisle or the entire stove. We

---

[2]https://github.com/facebookresearch/Detic
[3]https://github.com/facebookresearch/segment-anything

| Method | Training | Perception | Grasp prediction | Placement prediction |
|---|---|---|---|---|
| TAX-Pose [2] | $16.5 \pm 1.3$ h | - | $0.02 \pm 0.01$ s | $0.02 \pm 0.01$ s |
| R-NDF [17] | - | - | $21.4 \pm 0.5$ s | $42.5 \pm 1.8$ s |
| IW (Ours) | - | $29.6 \pm 0.2$ s | $0.01 \pm 0.01$ s | $0.003 \pm 0.004$ s |

Table 3: Approximate training and inference times for our method and baselines measured over five trials. R-NDF and IW do not have an explicit training phase, as they use demonstrations nonparametrically during inference. Only IW has a perception step that is separate from the action prediction step. We do not include the time it takes to capture a point cloud or to move the robot. Training and inference times were measured on a system with a single NVIDIA RTX 2080Ti GPU and an Intel i7-9700K CPU.

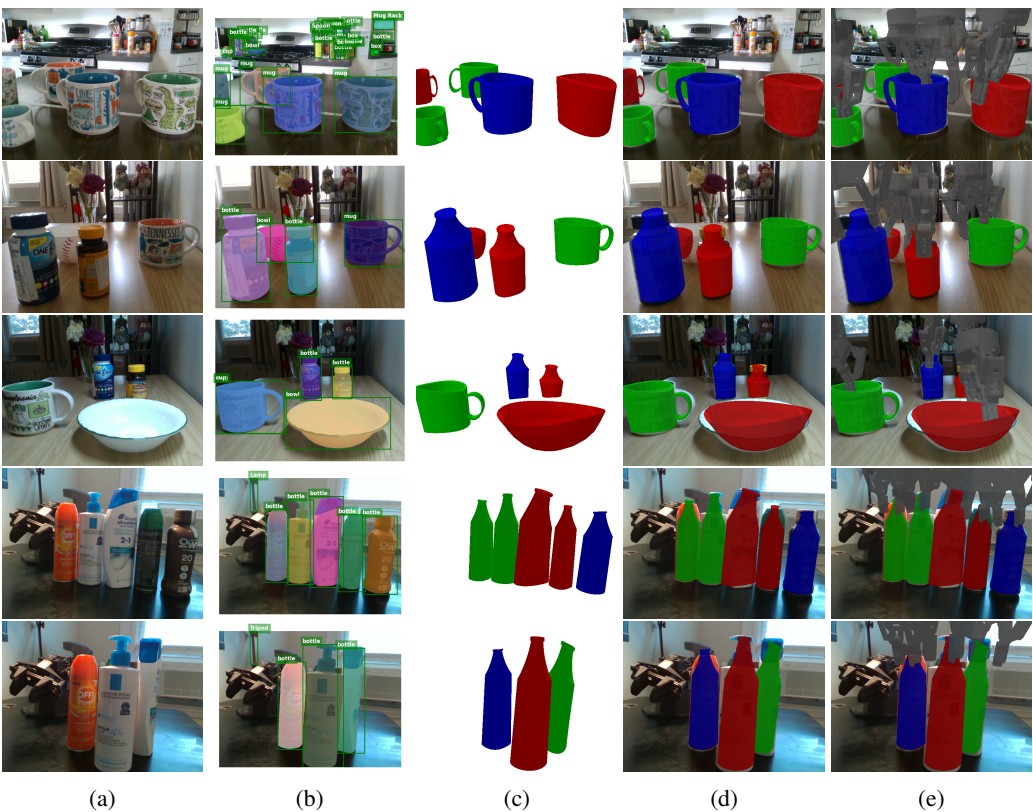

| (a) | (b) | (c) | (d) | (e) |
|---|---|---|---|---|

Figure 9: Additional examples, please see Figure 7.

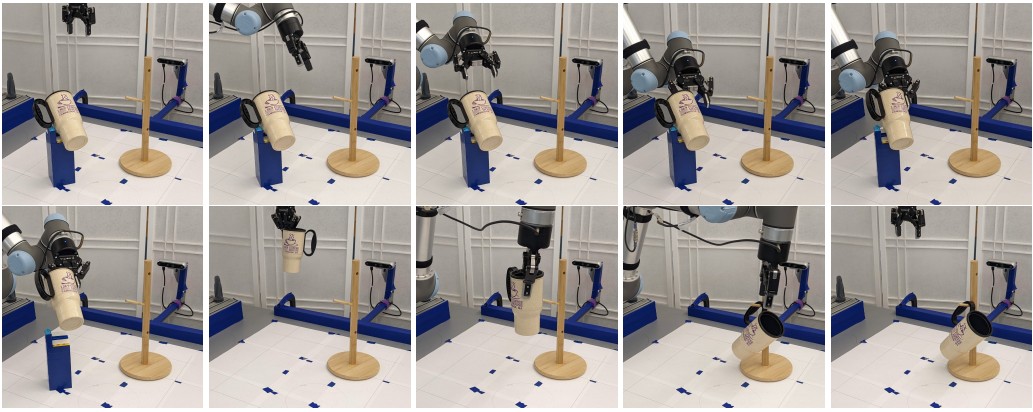

Figure 10: Example of mug on tree episode.

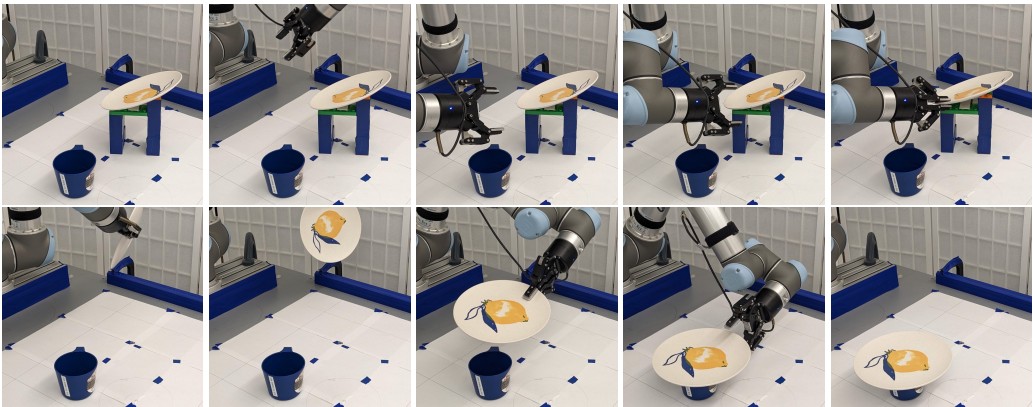

Figure 11: Example of bowl/plate on mug episode.

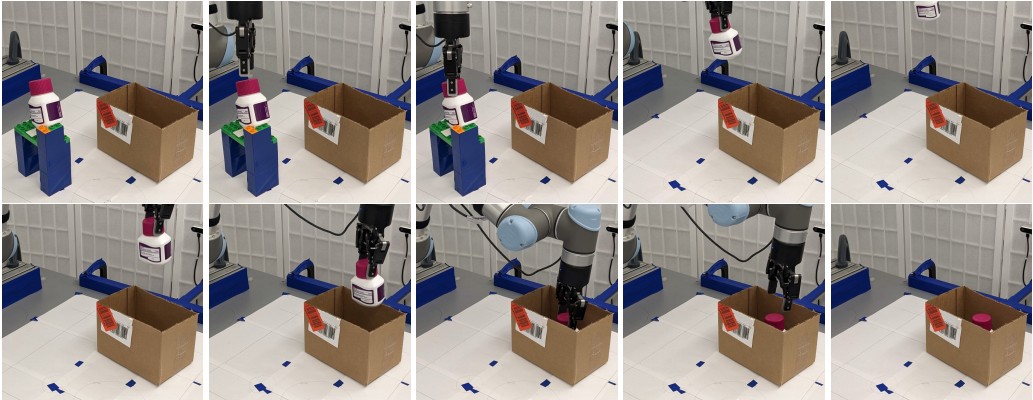

Figure 12: Example of bottle in box episode.

would have to manually crop the point cloud only to the top of the stove or the particular burner we want to place a pan onto.

**Limitations of joint shape and pose inference:** Joint shape and pose inference is prone to getting stuck in local minima. For example, instead of rotating a mug to match its handle to the observed point cloud, our inference method might change the shape of the mug to make the handle very small. We address this problem by using many random starting orientations – the full inference process takes 25 seconds per object on an NVIDIA RTX 2080 Ti GPU.

Pose inference might also fail when we do not see the bottom of the object. We subtract the table from the point cloud, so an observed point cloud of a mug might have an opening both at the top and at the bottom. Then, the inference process might not be able to tell if the mug is right side up or upside down.

