# OpenReview forum: "One-shot Imitation Learning via Interaction Warping"
_robot-learning.org/CoRL/2023/Conference — CoRL 2023 Poster_

### Official Review · Reviewer_M4iE · 2023-07-05

**Confidence:** 5
**Originality:** Very Good
**Technical Quality:** Excellent
**Clarity Of Presentation:** Very Good
**Impact:** 3

**Recommendation:**

Weak Accept: I recommend accepting the paper, but will not argue for my recommendation if the majority of other reviewers have a different opinion.

**Review:**

## Strengths:

The evaluation is good – a good choice of baseline methods is chosen, and the experimental design is appropriate. The presentation of the work is clear and well-structured, with well-designed figures. The method is original and clearly better than the baselines.

Weaknesses:
- If my understanding of the work is correct, the biggest weakness of this approach is that parts of the method appear heuristic/arbitrarily focused on the mug on rack task  - such as the choice of how virtual points are determined; or choosing to use the mug points before release from the gripper. This limits the applicability of the method to other use-cases.

- (L93) A full point cloud of the objects is necessary for the demonstrations, which may be difficult to obtain.

- The method is limited to inferring on objects that are similar to the ones in the demonstration set, and it would likely fail when dealing with objects that have additional parts, or missing parts - even if these are not relevant to the task

- The limitations section contains three limitations – the difficulty of using real RGB-D cameras along with segmentation, the 25 second inference time, and PCA being too simplistic for capturing of details. I expected a much more in-depth analysis of failure modes, use-cases in which the model would not be applicable, etc.

## Minor Comments for paper improvements that could be considered:

- Phi(.) from CPD can be described in more details.

- L147 typo “that are ignore”

- Figure 4. typo: “k nearest neighbours”

- L211 type “three RGB-D [cameras/sensors/images].”

- 4.2 could use a figure to illustrate the problem

- Tekden et al.,  “Affordance Transfer based on Self-Aligning Implicit Representations of Local Surfaces” appears to be doing something similar and might provide inspiration for a potential continuation of this method toward using object parts instead of full objects for this purpose.

- L255: single desktop GPU is less informative than a specific model GPU.


**Quality Of The Limitations Section:**

Limitations are not well addressed

**Questions For Rebuttal:**

-  Was at any point a real-world demonstration used, or was real world data only used for inference tests?

- L125-L128 are a bit unclear to me, why would it be large without the regularizer?

- Is there any explanation for the big drop in performance on upright mugs for 10 demonstrations (Table 1)? Typically more demonstrations are expected to produce better results. Similarly unexpected drops are observed in some of the other cells as well. This might indicate that the number of trials was not sufficient?

- Did you consider transferring grasps to novel grippers. Would it be possible with this method

**Robotics Focus:**

Sufficient demonstration on hardware

**Summary Of Paper:**

The work proposes the use of Coherent Point Drift as opposed to direct point correspondence, in order to find the semantically corresponding SO(3) manipulation action to a single demonstration. The method outperforms all baselines in several tasks, and its application is successfully demonstrated with a real robot.

**Summary Of Recommendation:**

My recommendation is weak acceptance since the work is interesting, novel and improving the state of the art, however, there are some more questions that need to be addressed. I accept because I am willing to argue for this paper’s acceptance. However, I’m also willing to argue the contrary based on any specifics that other reviews point out and can ultimately support either decision.

---

### Official Review · Reviewer_mU8N · 2023-07-09

**Confidence:** 3
**Originality:** Very Good
**Technical Quality:** Very Good
**Clarity Of Presentation:** Very Good
**Impact:** 4

**Recommendation:**

Strong Accept: I recommend accepting the paper and will argue for my recommendation even if other reviewers hold a different opinion.

**Review:**

The paper is interesting and quite clear, with supplementary material to clear up additional questions. The originality and significance of the paper are considerable.

**Quality Of The Limitations Section:**

Additional details required

**Questions For Rebuttal:**

the paper is interesting and the results look promising. As the authors point out themsleves, they use PCA model of shape warps, when something with higher capacity might work better. Also, the objects are somewhat simple, would it work on momre complex object (i.e. chair, even though a chair is somewhat bigger and more difficult to manipulate)?

**Robotics Focus:**

Sufficient demonstration on hardware

**Summary Of Paper:**

The paper introduces a method for one-shot learning of SE(3) robotic manipulation policies called Interaction warping it works such that for an object class it has one demonstration. The shape and contact points are warped to a different shape in the same class, therefore, with only one method you can theoretically, if all works well, manipulate all objects of that class. Results show improvement over other methods from the literature

**Summary Of Recommendation:**

Nice conference paper with potentially a very impactful and useful method.

---

### Official Review · Reviewer_odM7 · 2023-07-11

**Confidence:** 5
**Originality:** Fair
**Technical Quality:** Very Good
**Clarity Of Presentation:** Good
**Impact:** 3

**Recommendation:**

Weak Accept: I recommend accepting the paper, but will not argue for my recommendation if the majority of other reviewers have a different opinion.

**Review:**

**Quality**: The idea of detecting contact points from a demonstration, warping them to novel object instances, and analytically solving for end-effector and object poses that realise these contact points is very appealing and intuitive. The majority of the paper is also technically sound. However, there are some claims that are not well supported. For example, the authors claim that one of the advantages of the proposed method over prior work is that their "method is agnostic to the use of neural networks" without providing a justification for why this is an advantage. Moreover, in the limitations section, the authors say that their generative model "cannot capture the details of objects" while many neural-network-based models can, making it seem as if the lack of neural networks in the proposed method is a disadvantage.

**Originality**: Transferring grasp poses using the CDP + PCA model has been done in [1], while gradient descent on both the pose and the shape was previously done in [2]. The authors acknowledge both of these papers in their related work section and extend the previous line of work to placing/hanging.

**Clarity**: The clarity of the paper could be improved, as currently, the paper raises some questions, including:
- Do all CAD models used to fit the generative model need to be pre-aligned?
- Is the demonstration provided in simulation (as it must be provided on one of the CAD models used to fit the generative model)?

**Strengths**:
- The method requires only a single demonstration.
- Learned policies generalise to novel object instances.

**Weaknesses**:
- The method requires prior object knowledge.
- The demonstration must be given in simulation.

**References**:

[1] D. Rodriguez, A. Di Guardo, A. Frisoli and S. Behnke, "Learning Postural Synergies for Categorical Grasping Through Shape Space Registration," 2018 IEEE-RAS 18th International Conference on Humanoid Robots (Humanoids), Beijing, China, 2018, pp. 270-276, doi: 10.1109/HUMANOIDS.2018.8625032.

[2] D. Rodriguez, C. Cogswell, S. Koo and S. Behnke, "Transferring Grasping Skills to Novel Instances by Latent Space Non-Rigid Registration," 2018 IEEE International Conference on Robotics and Automation (ICRA), Brisbane, QLD, Australia, 2018, pp. 4229-4236, doi: 10.1109/ICRA.2018.8461169.


**Quality Of The Limitations Section:**

Limitations are not well addressed

**Questions For Rebuttal:**

- Do all CAD models used to fit the generative model need to be pre-aligned?
- Is the demonstration provided in simulation (as it must be provided on one of the CAD models used to fit the generative model)?
- Is there a way to provide the demonstration in the real world without having access to the object CAD model?
- How does the proposed method change when using more than one demonstration, as done in the experiments section?

**Robotics Focus:**

Sufficient demonstration on hardware

**Summary Of Paper:**

The authors propose Interaction Warping, a method for learning policies from a single demonstration for pick-and-place tasks involving two objects. The proposed method assumes prior knowledge and access to several CAD models from each object category. The authors use CPD with PCA to create a generative model for point cloud completion and warping. Given a single demonstration on one of the CAD models comprising the dataset used to fit the generative model, Interaction Warping first detects the contact points between the gripper and the object to be picked, and between the grasped object and the object it should be placed/hung on. At test time, the partial point cloud of both test objects is compleated and warped to the point clouds used during the demonstration using the learned generative model. The output warping is used to map the contact points retrieved from the demonstration to the novel object instances, which are then realised to complete the task.

**Summary Of Recommendation:**

Although the proposed method is interesting and appealing, the novelty appears to be incremental compared to prior work, introducing a method for transferring placement/hanging contact points that is analogous to the method used for transferring grasping contact points. Moreover, from my understanding, a major limitation of the method is that all CAD models from a single object category have to be pre-aligned, and that the demonstration must be provided in simulation on one of the models used to fit the generative model. Moreover, the paper could be better organised and its clarity improved.

---

### Official Review · Reviewer_XXj1 · 2023-07-18

**Confidence:** 3
**Originality:** Good
**Technical Quality:** Good
**Clarity Of Presentation:** Very Good
**Impact:** 4

**Recommendation:**

Weak Accept: I recommend accepting the paper, but will not argue for my recommendation if the majority of other reviewers have a different opinion.

**Review:**

*quality*

Pros
- Using shape warping to align point cloud is novel for one-shot imitation learning, and overall a neat idea.
- Empirical analysis includes two baseline and 3 sim/real tasks. 100 trials are conducted for each task*method combination, making the results significant and convincing.

Cons
- It is not clear if comparing the proposed method with Neural Descriptor Field (NDF) baselines is a fair comparison. The proposed method requires access to "a set of category-level generative object models" (line 112, page 3), which I assume requires additional training data. While NDF baselines may not be able to make use of this additional information.
- It is not clear how general the proposed method is. Similar to previous works, the method is tested only on cup/bowl-like objects. The procedure to extract "nearby points" (line 175, page 6) appears to be quite specific to the Mug Tree task.

*clarity*
- The presentation is clear and detailed.

*originality*
- The proposed method is original and novel.

*Significance*
- The paper is interesting and significant.





**Quality Of The Limitations Section:**

Additional details required

**Questions For Rebuttal:**

For Con-1:
- More details about the data used for training the generative model/PCA should be provided.
- Could NDF baselines leverage these data? Is the comparison fair?

For Con-2
- More discussions about which object categories and tasks will be suitable for this approach.
- What are some example categories that warping methods may struggle? Does NDF approaches have similar issues? What are some object categories that both warping and NDF would struggle?
- This would give readers who are less familiar with this line of work better intuition.

**Robotics Focus:**

Sufficient demonstration on hardware

**Summary Of Paper:**

The paper proposes a new method, interaction warping, to tackle one-shot imitation learning. The overall pipeline uses keypoints on the object point cloud to predict grasps and other robot actions, and such keypoints are obtained through collision checking and other heuristics.

The main novelty of this work lies in how keypoints detected in the training object can be effectively transferred to unseen objects in the same category. Unlike previous works with descriptor fields, this paper uses shape warping, a way to align a new object to a canonical object through individual point displacements. This approach is shown to achieve higher performance when compared to previous methods.

**Summary Of Recommendation:**

The paper is novel and interesting, though some flaws need to be addressed.

---

### Author Response · Authors · 2023-08-09
**General comment**

We thank the reviewers for their comprehensive feedback. We are pleased the reviews found our application of shape warping to one-shot imitation learning original (XXj1, M4iE), intuitive (odM7) and impactful (mU8N) and the presentation of our paper clear (XXj1, mU8N, M4iE) and well-structured (M4iE).

We would like to clarify a major point (Reviewers odM7, M4iE):
**Our demonstrations (in real world experiments) are collected on the physical robot using teleoperation. We do not know the CAD model of the objects used in the demonstrations, only their classes. We do not have the CAD model of any object used in real world experiments (we collected them randomly and did not scan them).**

For example, Reviewer odM7 states: “a major limitation of the method is [...] that the demonstration must be provided in simulation on one of the models used to fit the generative model”. This is untrue, as we provide demonstrations in the real world with arbitrary objects. We believe this misconception was caused by the images we used in Figure 3 and we updated the figure in our revision.

Further, Reviewers XXj1 and mU8N asked for more intuition about object classes suitable for shape warping and Reviewers XXj1, odM7 and M4iE asked several questions about the datasets we used. We updated the experiments section and added an extended limitation section in Appendix D to answer these questions.

We uploaded a revised version of our paper to each of our responses below.

---

### Decision · Program_Chairs · 2023-08-30

**Decision:**

Accept (Poster)

**Comment:**

This paper introduces a new method for one-shot imitation learning, which proposes the use of Coherent Point Drift to align different object instances from the same object category, based on point cloud observations. Heuristics used to detect keypoints from a demonstration then allow the demonstration to be transferred to novel objects of the same category. Simulation and real-world experiments show that this method outperforms some recent well-known baselines.

Reviewer scores before the rebuttal were 1 x “weak reject”, 2 x “weak accept”, and 1 x “strong accept”. Authors provided a rebuttal, and an updated paper which included some minor clarifications to the text. Following the rebuttal, reviewer scores were 3 x “weak accept”, and 1 x “strong accept”. The “weak reject” reviewer upgraded to “weak accept” following satisfactory answers to their questions. The AC then initiated a discussion amongst reviewers, although there was already a positive consensus amongst the reviewers and so the discussion was short.

Overall, the AC and reviewers find this to be an interesting and sensible method, with compelling results. Some general criticisms still remain, such as (1) the need for aligned CAD models for each object category, (2) the use of a heuristic for the virtual points which may not be appropriate for other tasks, and (3) the limited range of tasks evaluated on. However, there is still a consensus that this paper is useful for the robot learning community and that it should therefore be accepted at CoRL. Please be sure that for the final paper, the clarifications added to the updated paper remain (the updated paper is currently over 8 pages), and that the three limitations above are adequately addressed.